# A Refractive Index Sensor Based on H-Shaped Photonic Crystal Fibers Coated with Ag-Graphene Layers

**DOI:** 10.3390/s20030741

**Published:** 2020-01-29

**Authors:** Tianshu Li, Lianqing Zhu, Xianchao Yang, Xiaoping Lou, Liandong Yu

**Affiliations:** 1School of Instrument Science and Opto-Electronics Engineering, Hefei University of Technology, Hefei 230009, China; litianshu@mail.hfut.edu.cn; 2Beijing Laboratory of Optical Fiber Sensing and System, Beijing Information Science & Technology University, Beijing 100016, China; yangxianchao@tju.edu.cn (X.Y.); louxiaoping@bistu.edu.cn (X.L.)

**Keywords:** photonic crystal fibers, surface plasmon resonance, H-shaped optical fiber, liquid refractive index, graphene

## Abstract

An Ag-graphene layers-coated H-shaped photonic crystal fiber (PCF) surface plasmon resonance (SPR) sensor with a U-shaped grooves open structure for refractive index (RI) sensing is proposed and numerically simulated by the finite element method (FEM). The designed sensor could solve the problems of air-holes material coating and analyte filling in PCF. Two big air-holes in the x-axis produce a birefringence phenomenon leading to the confinement loss and sensitivity of x-polarized light being much stronger than y-polarized. Graphene is deposited on the layer of silver in the grooves; its high surface to volume ratio and rich π conjugation make it a suitable dielectric layer for sensing. The effect of structure parameters such as air-holes size, U-shaped grooves depth, thickness of the silver layer and number of graphene layers on the sensing performance of the proposed sensor are numerical simulated. A large analyte RI range from 1.33 to 1.41 is calculated and the highest wavelength sensitivity is 12,600 nm/RIU. In the linear RI sensing region of 1.33 to 1.36; the average wavelength sensitivity we obtained can reach 2770 nm/RIU with a resolution of 3.61 × 10^−5^ RIU. This work provides a reference for developing a high-sensitivity; multi-parameter measurement sensor potentially useful for water pollution monitoring and biosensing in the future.

## 1. Introduction

Surface plasmon resonance (SPR) is considered one of the most promising sensing technologies in environmental protection [1,2], chemical reaction monitiring [3], biomedical sensing [4] and other photonic devices [5,6] due to its advantages being of label-free, accurate and of high sensitivity. SPR occurs when an evanescent wave matches free electrons on a metal surface. A narrow absorption peak can be obtained at the resonance wavelength, which is very sensitive to any changes of the refractive index (RI) of the medium. The earliest SPR sensing devices were prism based [7,8], such as the Kretschmann–Raether prism, which can realize a detection limit up to 10^−6^ RIU. Even though the prism can achieve high sensitivity, it has the disadvantage of costly integration, limited mechanical reliability and difficulties in mass production [9]. Optical fiber-based SPR sensors have been studied by researchers in recent years due to their advantages of easy miniaturization, remote sensing and in-situ monitoring [10]. In most fiber-based SPR sensors, the fiber is tapered or side polished and the exposed part is coated with a noble metal layer to produce SPR [11]. Gold and silver are the most common metals used in sensing layers. Gold is the most suitable plasmonic material because of its high biocompatibility and chemical stability when exposed to an aqueous and humid environment. Silver shows a sharper resonance peak compared to gold, hence it has higher detection accuracy, but it is prone to oxidation. Graphene has a π orbitals forming a dense cloud that could block the gap within its atomic rings. This results in a repelling field which can prevent metal surface oxidation [12,13]. Graphene also has the advantage of a high surface to volume ratio, and broad band optical and plasmonic properties, so many researchers prefer to use this material above the metal layer in SPR sensors [14,15]. 

Photonic crystal fibers (PCF) have advantages such as minimized size, tunable geometric parameters and birefringence phenomena, which make them a suitable choice for SPR-based sensors. Moreover, by changing the geometric parameters of the PCF structure, the effective RI of its core-guided mode can phase match the plasmon mode. Hassani and Skorobogatiy [16] firstly combined a microstructure optical fiber with microfluidics, and the effective RI of the core-guide mode could be tailored by changing the geometrical parameters of the structure, which helps in phase matching of core-guided mode and plasmon mode. Three years later Yu et al. [17] designed a microstructure fiber sensor with selectively metal-coated air holes to replace the full air holes-coated sensor. This approach will not only reduce the gold consumption by half, but also has the potential to reduce the material absorption losses at other wavelengths where there is no resonance effect. In the above- mentioned sensors, the analyte needs to be filled inside the voids of the fiber, and the metal layer fabrication process is challenging. In recent years, many researchers have employed the D-shaped PCF with metal and analyte directly deposited on the exposed section. Erdmanis et al. [18] presented an H-shaped SPR optical fiber sensor that can achieve a wavelength sensitivity of 5000 nm/RIU. Dash et al. [4] proposed a graphene-Ag-coated D-shaped PCF-SPR sensor with birefringence phenomenon, which wavelength sensitivity can reach 3700 nm/RIU with a resolution of 2.7 × 10^−5^ RIU. In 2018, Luan et al. [19] designed a D-shaped PCF with a laterally accessible hollow-core. A side-opening channel is introduced to allow the analyte to gain access into the hollow-core (the sensing region), which also provides a possibility for real time sensing. For these external approaches, the air-holes in the PCF are no longer used as microfluidic channels for the sensing application [20].

In this paper, we design an H-shaped PCF-SPR sensor to detect analyte RI. Two symmetrical U-shaped grooves are punched through the second layer of the air-hole on the y-axis. The open grooves are directly coated with Ag-graphene layers, which simplifies the preparation process of the sensing layer coating, meanwhile eliminating the need to inject the analyte in advance. The asymmetrical structure leads to a birefringence phenomenon whereby the confinement loss and sensitivity of x-polarized light is much stronger than for y-polarized. The sensor performance can be numerically analyzed by changing the air-hole size and the thickness of the material layers. An analyte RI range from 1.33 to 1.41 is calculated, and the average wavelength sensitivity we obtained is 2770 nm/RIU with a resolution of 3.61 × 10^−5^ RIU in a linear RI range of 1.33–1.36.

## 2. Structure Design and Simulated Modeling

The H-shaped PCF-based SPR sensor designed in this paper is shown in Figure 1. The PCF consists of two layer air-holes in a hexagonal layout. The center air-hole can lower the effective RI of core-guided mode to be in phase with the plasmon mode. There are two big air-holes in the first layer, leading to a strong birefringence phenomenon and coupling of a particularly polarized light with the metal dielectric interface. Two symmetrical U-shaped grooves can be fabricated by a femtosecond laser micromachining technique [21] or using the hydrofluoric acid solution etching method [22]. Silver is used as plasmonic material for coating on the grooves surface so as to generate an ample amount of plasmonic effect. Graphene is deposited on the silver layer to protect it from oxidation and control the plasmon mode. The silver layer coating can be realized by the chemical vapor deposition technique [23] and for the graphene coating procedure we refer to [24].

The parameters of the designed structure are *Λ* = 2 μm, *d_c_/Λ* = 0.5, *d_1_/Λ* = 0.5, *d_2_/Λ* = 0.9, *d_3_/Λ* = 0.8, *t_Ag_* = 40 nm and *t_graphene_* = 4.08 nm, where *Λ* is the gap between two air-holes, *d_c_*, *d_1_*, *d_2_*, *d_3_* are diameter of the air-holes in the PCF shown in Figure 1, *t_Ag_* is the thickness of the silver layer and *t_graphene_* is the thickness of the graphene layer. The PCF structure can be numerically simulated by the finite element method (FEM) using COMSOL Multiphysics software. We use perfectly matched layers (PML) [25] as the boundary condition and the total number of mesh elements is 561,069. The simulation for modal analysis is done in the XY plane and the light propagation is along the Z direction. The fiber material is fused silica and the RI is determined using the Sellmeier equation [26]:(1)$$n^{2}(\lambda )=1+\frac{A_{1}\lambda^{2}}{\lambda^{2}-B_{1}}+\frac{A_{2}\lambda^{2}}{\lambda^{2}-B_{2}}+\frac{A_{3}\lambda^{2}}{\lambda^{2}-B_{3}}$$
where *A*_1_ = 0.696166300, *A*_2_ = 0.407942600, *A*_3_ = 0.897479400, *B*_1_ = 4.67914826 × 10^−3^ μm^2^, *B*_2_ = 1.35120631 × 10^−2^ μm^2^ and *B*_3_ = 97.9340025 μm^2^. The RI of graphene can be calculated by the following equation [14]:(2)$$n_{g}=3+\frac{i\times 5.446\mu m^{-1}\times \lambda }{3}$$
where *λ* is the vacuum wavelength. In Figure 1
*t_graphene_* = 4.08 nm, the number of graphene layers is *L* = 12. Each single layer of graphene has a thickness of 0.34 nm. The RI of silver can be calculated by the Lorentz-Drude model.

## 3. Results

### 3.1. Birefringence Phenomenon

Asymmetrical structure leads to a birefringence phenomenon. The dispersion relations and electric field distribution of the core-guided mode and the plasmon mode are shown in Figure 2 with the analyte RI 1.33. The blue Gaussian-like solid curve represents the imaginary pars of the effective index of x-polarized core-guided mode with wavelength of the incident light, and the blue dotted curve represents the imaginary pars of the effective index of the y-polarized core-guided mode. The black solid line and dotted line represent the real parts of the effective RI of x-polarized core-guided mode and the y-polarized mode, respectively. The red line represents the plasmon mode. Resonance occurs when the real parts of the core-guided mode matches with the plasmon mode at a specific wavelength. Points (c) and (d) shown in Figure 2 represent the electric field distribution of the resonance position in the x-polarized and y-polarized directions, respectively.

Confinement loss is used to estimate the sensor performance and can be defined as follows [27]:(3)aloss(dBm)=8.686×2πλ×Im[neff]
where *λ* is the operating wavelength and Im[*n_eff_*] is the imaginary part of the effective RI of the core-guided mode. 

Figure 3a shows loss spectra of x-polarized and y-polarized mode when the analyte RI varies from 1.33 to 1.34. The plasmon mode is affected by the RI of the analyte liquid, so that changes the phase matching wavelength between the core mode and plasmon mode. As the RI of the liquid increases, the resonance wavelength will shift to a longer wavelength direction. We can calculate that the confinement loss of x-polarized mode is 1.795 × 10^2^ dB/cm, while the y-polarized peak is 1.123 × 10 dB/cm. Wavelength sensitivity and amplitude sensitivity can be used to evaluate sensor performance. They are defined as follows [28]:(4)$$S_{\lambda }(nm/RIU)=\frac{\partial \lambda_{peak}}{\partial n_{a}}$$
(5)$$S_{A}(RIU^{-1})=\frac{1}{a(\lambda ,n_{a})}\times \frac{\partial a(\lambda ,n_{a})}{\partial n_{a}}$$
where *S_λ_* is wavelength sensitivity, *∂λ_peak_* is the peak wavelength shift and *∂n_a_* is the variation of analyte RI. *S_A_* is the amplitude sensitivity and *α (λ, n_a_)* is the confinement loss in the case *n_a_* = 1.33. When the RI of the analyte varies from 1.33 to 1.34, the x-polarized peak has a higher wavelength sensitivity (2300 nm/RIU) than the y-polarized one (1900 nm/RIU). Figure 3b shows the relationship between amplitude sensitivity and wavelength, conirming the the x-polarized peak has a higher amplitude sensitivity (34.64 RIU^−1^) than the y-polarized one (23.56 RIU^−1^). For SPR-based RI optical fiber sensors, the x-polarized peak is more suitable than the y-polarized one, and the birefringence phenomenon can be used in multi-parameter measurements in the future.

### 3.2. RI Sensitivity of the Sensor

An analyte RI range from 1.33 to 1.41 is calculated to investigate the sensing performance of the designed sensor. Figure 4a shows the loss spectra variation at the RI range of 1.33 to 1.35 in steps of 0.005. As the RI increaes, the resonance wavelength shifts to the longer wavelength direction and the peak loss increases. As shown in Figure 4b, the fitting line is divided into two parts and RI = 1.36 is regarded as a turning point. The red line at the RI range of 1.33–1.36 has an average wavelength sensitivity of 2770 nm/RIU, with the corresponding *R*^2^ value of 0.9947. The blue line at the RI range of 1.36–1.41 has an average wavelength sensitivity of 6057 nm/RIU, with the corresponding *R*^2^ value of 0.9555. As analyte RI increases from 1.33 to 1.41, the effective RI of the plasmon mode becomes more and more close to the core-guided mode (≈1.45), and the phase matching becomes easier, so that the confinement loss will increase and achieve higher sensitivity. The highest wavelength sensitivity is 12,600 nm/RIU at 1.41. If the spectrograph resolution is 0.1 nm, the minimum resolution of the sensor is 7.94 × 10^−6^ RIU. Compared with other similar works, the designed sensor exhibits comparable sensitivity and resolution, as shown in Table 1.

### 3.3. Investigation of Various Structural Parameters

The influence of structure parameters of H-shaped PCF such as air-holes size, U-shaped grooves depth, the thickness of silver layer and number of graphene layer on the sensing performance of the proposed sensor can be numerically simulated.

As shown in Figure 5a, the resonance wavelength red-shifts slowly and the confinement loss increases obviously with the *d_c_/Λ* change from 0.3 to 0.6. The core-guide mode can be influenced by the size of the center air-hole, and a larger air hole will lower the effective RI of core-guided mode and make it closer to the analyte RI. When the RI changes from 1.33 to 1.34, with the increase of *d_c_*, the amplitude sensitivity has a slightly raise, to 33.31, 33.88, 34.64 and 35.75 RIU^−1^ respectively. The wavelength is also increased according to the Figure 5b. When *d_c_/Λ* = 0.3, 0.4, 0.5 and 0.6, the wavelength sensitivities are 2100, 2200, 2300 and 2500 nm/RIU, but as described in [31], when the *d_c_* is too large, the incident light will not be confined well in the core, the mode coupling will weaken and the peak loss will decrease, so we choose *d_c_/Λ* = 0.5 in this work.

Keeping other parameters unchanged, the results when the *d_3_* diameter increases from 0.6 to 0.8 are shown in Figure 6. These two air-holes in the first layer represent the asymmetrical structure of the sensor, which can affect the birefringence significantly. According to the results shown in Figure 6a,b, when *d_3_/Λ* increase from 0.6 to 0.8, the resonance wavelength shifts from 611 to 615 nm, while the amplitude remains nearly unchanged.

The open structure of the H-shaped fiber has two symmetrical U-shaped grooves, which are fabricated from the edge of the fiber to the center of the second layer air-holes in the y-axis direction. The original depth of *H* is from the center of air-hole to the edge of fiber, then we also analyze the effect of an *H* depth decrease to 0.3 and 0.5 μm. As shown in Figure 6c, the confinement loss will decrease when *H* increases, and the resonance wavelength shifts to the shorter wavelength with the grooves closer to the fiber core. The polarization characteristics can be affected by the depth of grooves, and as shown in Figure 6d, the amplitude sensitivity decreases slightly when *H* increases.

The material layer thickness could affect the plasmon mode and thus change the sensor’s performance. Figure 7a,b depicts the variation in the loss peak with the thickness of the silver layer, keeping *L* = 12, *d_3_/Λ* = 0.8, *dc/Λ* = 0.5. When the silver layer thickness increases from 20 nm to 50 nm, the loss spectra dampens obviously and shifts to a longer wavelength, and the amplitude also decreases from 56.37 RIU^−1^ to 24.54 RIU^−1^ as shown in Figure 7b. Part of the core-guided mode is used to overcome the damping loss, hence the interaction between the core-guided mode and the plasmon mode decreases. The corresponding wavelength sensitivities are 1800, 2200, 2300 and 2600 nm/RIU as *t_Ag_* increases from 20 nm to 50 nm. We can find that the amplitude sensitivity and wavelength sensitivity present opposite changes, which should be optimized before using the sensor in practice.

Figure 7c,d shows the variation in loss peaks when the number of graphene layers (*L*) changes from nine to fifteen (*L* = 9 to *L* = 15). Other parameters such as *t_Ag_* = 40, *d_3_/Λ* = 0.8, *dc/Λ* = 0.5 remain constant. The resonance peak shifts to the longer wavelength and becomes broadened with an increase in the number of graphene layers. The peak broadening is caused by the imaginary part of the dielectric constant of graphene dominance, so that the damping loss increases and less energy couples to the plasmon mode. Like with the silver layer, the amplitude sensitivity will decrease from 46.42 to 27.12 RIU^−1^ as the graphene becomes thicker, and the wavelength sensitivity increases from 2200 to 2400 nm/RIU. A thinner graphene layer can provide a shaper resonance peak but lower wavelength sensitivity, so in this paper we choose *L* = 12 (*t_graphene_* = 4.08 nm).

Considering the difficulty to control the thicknesses of silver and graphene layers precisely during fabrication, the influence of manufacturing tolerance is analyzed in detail. As shown in Figure 7a–d, a 10% tolerance of the silver layer thickness will lead to the wavelength sensitivity increasing 53 nm/RIU and the amplitude sensitivity decreasing by 2 RIU^−1^. Similarly, a 10% tolerance of the graphene layer thickness will lead to the wavelength sensitivity increasing 30 nm/RIU and the amplitude sensitivity decreasing by 2.9 RIU^−1^. Thus we can know that the sensitivity caused by the silver layer thickness is much stronger than that caused by the graphene layer thickness, and the wavelength sensitivity variation is much larger than that of amplitude.

## 4. Conclusions

An Ag-graphene SPR-based H-shaped PCF RI sensor has been numerically simulated by the finite element method (FEM). This structure realizes the coupling of plasmon mode and core-guided mode. When SPR occurs, parts of the light will be transferred from the core of the fiber to the metal-dielectric interface so as to realize analyte liquid RI sensing. A birefringence phenomenon leads to the loss peak in x-polarized mode being higher than in y-polarized mode, which can be used in multi-parameter measurement. The U-shaped grooves open structure we propose simplifies the preparation process of the material layer, while eliminating the need to inject the analyte in advance. The wavelength and amplitude sensitivity can be affected by changing the structure of the PCF. When evaluating the performance of the sensor in practice, the thickness of the coating material should be selected reasonably. The wavelength sensitivity we obtained is as high as 12,600 nm/RIU with a resolution of 7.94 × 10^−6^ RIU. This structure provides a reference for the development a high-sensitivity, multi-parameter measurement sensors for use in water pollution monitoring and bio-sensing.

## Figures and Tables

**Figure 1 sensors-20-00741-f001:**
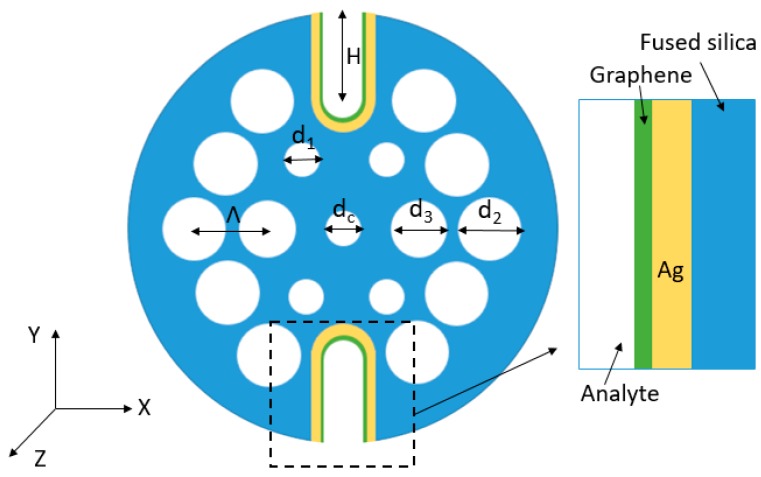
Schematic of the designed Ag-Graphene coated PCF-SPR sensor.

**Figure 2 sensors-20-00741-f002:**
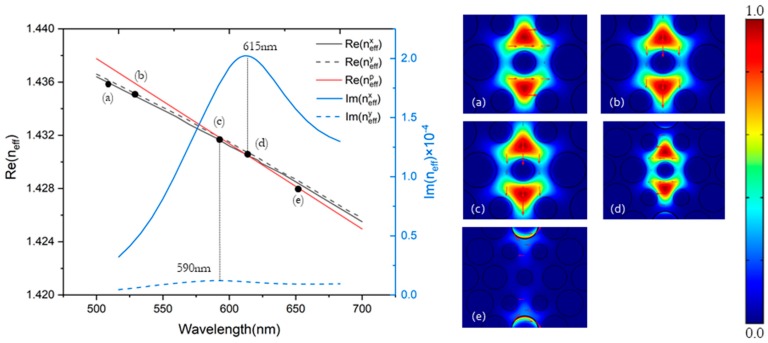
Dispersion relations and electric field distribution of core-guided modes and the plasmon mode with *n_a_* = 1.33. (**a**) x-polarized core mode at *λ* = 500 nm; (**b**) y-polarized core mode at *λ* = 500 nm; (**c**) y-polarized core mode at *λ* = 590 nm (phase matching point); (**d**) x-polarized core mode at *λ* = 615 nm (phase matching point); (**e**) Plasmon mode at *λ* = 650 nm.

**Figure 3 sensors-20-00741-f003:**
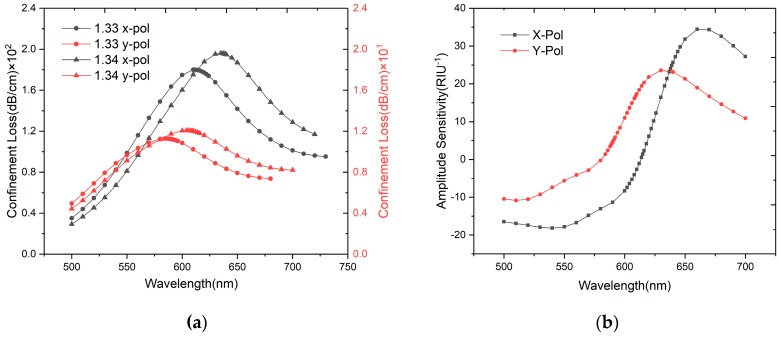
(**a**) Loss spectra of x-polarized and y-polarized with analyte RI = 1.33 and 1.34; (**b**) Amplitude sensitivity of x-polarized and y-polarized.

**Figure 4 sensors-20-00741-f004:**
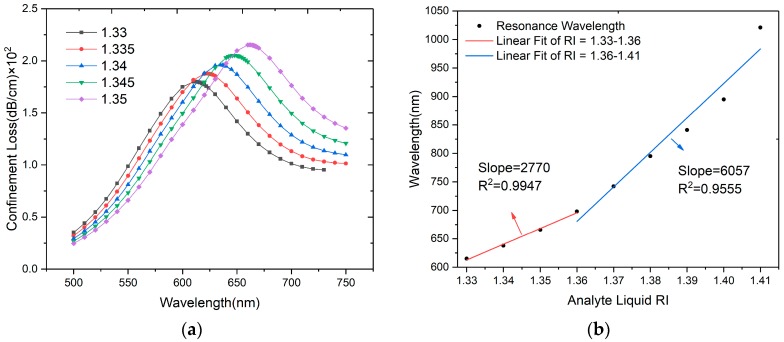
(**a**) Loss spectra of the designed sensor when *n_a_* varies from 1.33 to 1.35; (**b**) Relationship between resonance wavelength and RI in the range of *n_a_* from 1.33 to 1.41.

**Figure 5 sensors-20-00741-f005:**
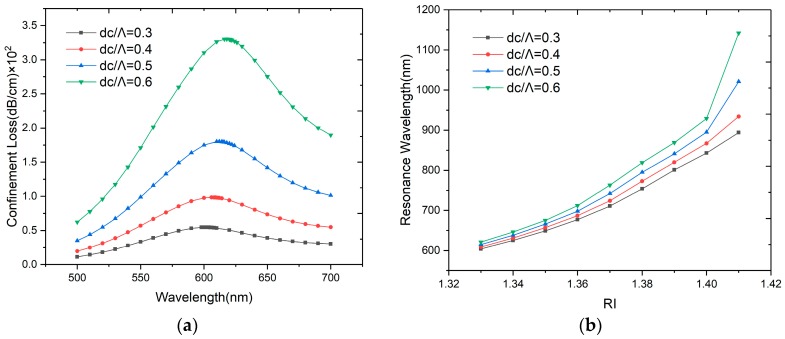
(**a**) Loss spectra of the designed sensor when *d_c_/Λ* = 0.3, 0.4, 0.5 and 0.6 with *n_a_* = 1.33; (**b**) Relationship between resonance wavelength and RI when *d_c_/Λ* = 0.3, 0.4, 0.5 and 0.6 in the range of *n_a_* from 1.33 to 1.41.

**Figure 6 sensors-20-00741-f006:**
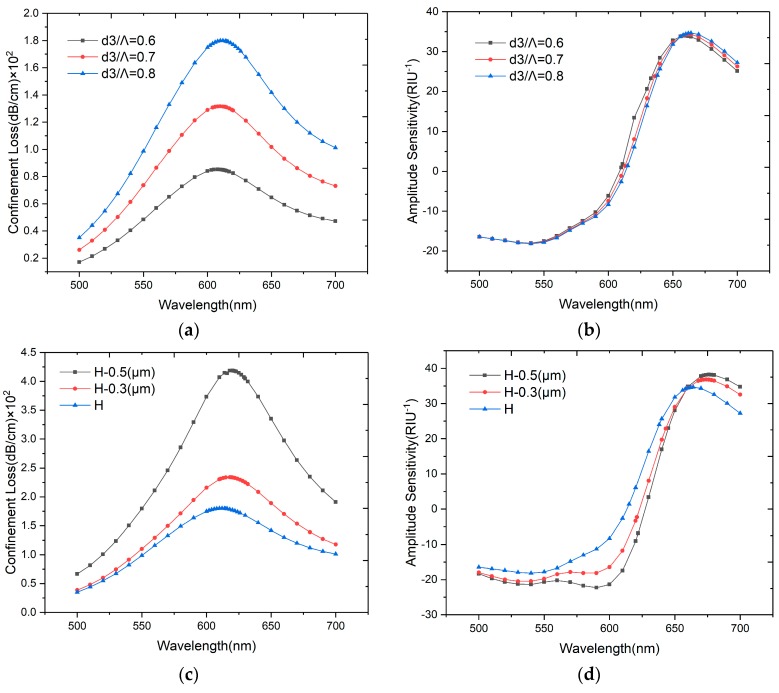
(**a**) Loss spectra of the designed sensor when *d_3_/Λ* = 0.6, 0.7 and 0.8 with *n_a_* = 1.33; (**b**) Amplitude sensitivities of the proposed sensor when *d_3_/Λ* = 0.6, 0.7, 0.8 with *n_a_* = 1.33; (**c**) Loss spectra of the designed sensor when *H* changed with *n_a_* = 1.33; (**d**) Amplitude sensitivities of the proposed sensor when *H* changed with *n_a_* = 1.33.

**Figure 7 sensors-20-00741-f007:**
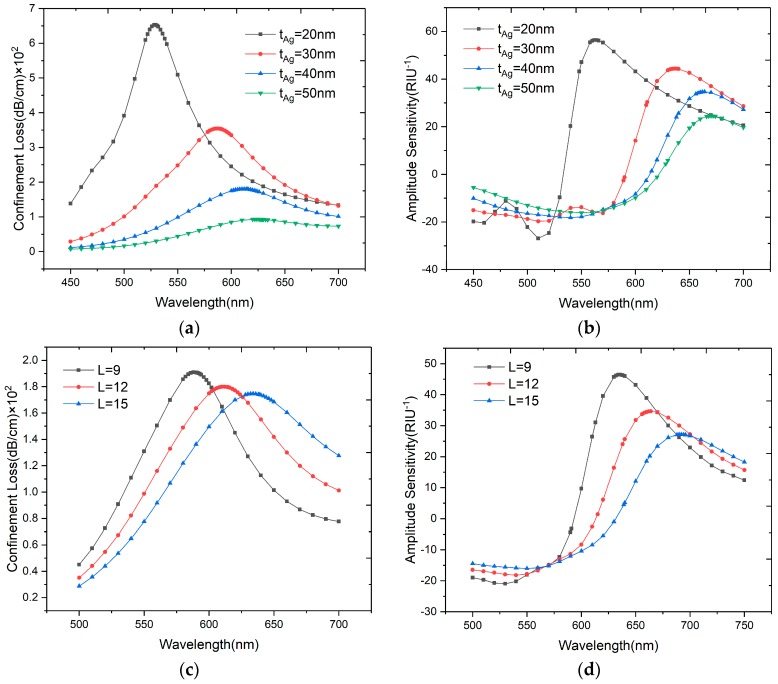
(**a**) Loss spectra of the designed sensor when *t_Ag_* = 20, 30, 40 and 50 nm with *n_a_* = 1.33; (**b**) Amplitude sensitivities of the proposed sensor when *t_Ag_* = 20, 30, 40 and 50nm with *n_a_* = 1.33; (**c**) Loss spectra of the designed sensor when *L* = 9, 12 and 15 with *n_a_* = 1.33; (**d**) Amplitude sensitivities of the proposed sensor when *L* = 9, 12 and 15 with *n_a_* = 1.33.

**Table 1 sensors-20-00741-t001:** Performances of other PCF-SPR sensors.

Characteristics	RI Range (RIU)	Wavelength Sensitivity (nm/RIU)	Resolution (RIU)	Ref.
Graphene-Ag coated D-shaped PCF sensor	1.33–1.37	3700	2.7 × 10^−5^	[4]
Exposed-core grapefruit fiber based SPR sensor	1.33–1.42	13500	7.41 × 10^−6^	[9]
Graphene-Ag coated outer surface of PCF	1.33–1.37	N/A	4 × 10^−5^	[14]
PCF-SPR sensor with enhanced microfluidics	1.33–1.34	N/A	~10^−6^	[16]
Selectively coated PCF-SPR sensor	1.33–1.41	5500	1.82 × 10^−5^	[17]
SPR sensor based on H-shaped fiber	1.32–1.33	5000	2 × 10^−5^	[18]
Au coated PCF with laterally accessible hollow-core	1.46–1.47	7200	1.38 × 10^−5^	[19]
Graphene-Ag coated PCF sensor	1.33–1.35	2520	3.97 × 10^−5^	[27]
Au coated dual core PCF sensor	1.33–1.51	6021	1.66 × 10^−5^	[29]
Au coated side-polished FMF-SPR	1.333–1.404	4903	2.04 × 10^−5^	[30]
SPR sensor based on exposed-core PCF	1.33–1.42	16400	6.1 × 10^−6^	[31]
This work	1.33–1.41	12600	7.94 × 10^−6^	N/A

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
