# Peer review of "A Refractive Index Sensor Based on H-Shaped Photonic Crystal Fibers Coated with Ag-Graphene Layers"

_sensors, 2020, doi:10.3390/s20030741_

Round 1

Reviewer 1 Report

In this manuscript, the authors studied the effect of stub-like defects in the photonic crystal fiber (PCF) for the sensing of the refractive index. The structure allows the surface plasmon resonance with the birefringence, increasing the degrees of freedom for sensing functionalities. Though the manuscript does not include novel physics or design scheme, the engineering and operation performance are rigorously analyzed using proper performance metrics. This manuscript provides the proper design guideline for sensing with the PCF platform. I thus suggest the acceptance of this manuscript.

Author Response

Dear reviewer,

We appreciate very much for the positive comments and affirmations. Thanks so much for agreeing to receive this paper.

Sincerely,

Tianshu Li

Hefei University of Technology

litianshu@mail.hfut.edu.cn

Reviewer 2 Report

In the manuscript "Refractive Index Sensor Based on H-Shaped Photonic Crystal Fiber Coated with Ag-Graphene layers" by Tianshu Li et al, an Ag-graphene layers coated H-shaped photonic crystal fiber (PCF) surface plasmon resonance (SPR) sensor, with U-shaped grooves open structure, is proposed and numerically simulated by finite element method (FEM) for refractive index (RI) sensing. This work is solid and can provides a reference for developing a high-sensitivity, multi-parameter measurement sensor used in water pollution monitoring and bio-sensing in future. In my opinion, this paper has significant merits for publication but some of the issues should be addressed prior to be accepted. 1. Whether the author can make the actual model based on the optimal parameters need to be tested on refractive index sensing performance. 2. In figure 4(a), can the author explain why the resonance wavelength will shift to the longer wavelength direction instead of the shorter with the RI of liquid increasing? 3. In the section of “3.2. Investigation of Various Structural Parameters”, There are some unit errors in the third part of the article. Such as“Same like silver layer, the amplitude sensitivity will decrease with graphene become thicker, and sensitivity increase from 2200 to 2400 nn/RIU”. 4. About “Surface plasmon resonance”, the latest references should be mentioned: Effect of slit width on surface plasmon resonance. Results in Physics 15 (2019) 102711; A numerical research of wideband solar absorber based on refractory metal from visible to near infrared. Optical Materials 2019, 97, 109400; 5. In conclusion, I think it could be accepted after all of my concerns above are properly addressed.

Author Response

Dear reviewer, the point-by-point response please see the attachment.

Reviewer 3 Report

The authors of this paper report a proposed new geometry fiber sensor and present numerical simulations of potential performance.  The sensor works on the principle of differential polarisation loss as a function of the refractive index of the analyte surrounding the fiber.

As a reviewer the paper is difficult to review as the English expression and grammar is of a very poor standard. This will require extensive editing.

The overall geometry proposed appears to have interesting predicted theoretical performance. The authors have presented the modelled electric fields as well as the wavelength dependent modal losses.

Could some details of the modelling package be included in the paper ? For instance is there anything unique in this work other than the proposed geometry ? What assumptions have the authors taken ? Is there any level of validation of this modelling technique using a standard fiber ?

My main concern is that without any experimental validation of this particular design, it is not clear if this design is realistic. I appreciate the ability to theoretically model any structure, however I am concerned that there is no discussion of why this proposed new design is feasible, or evidence that it could in fact be fabricated.

I note that the fiber diameter is not given, if it is a standard 125 microns the required groove is quite narrow and deep which would be a multi-step fabrication challenge.

For instance:

While it is possible to femtosecond micromachine notches into the fibre, this will result in surface structure – how can realistic surface structure be included in the model ? The silver and graphene layers are highly specified at 40 nm and 4.08 nm, respectively – this does not appear practical. How can such layers be achieved (or even measured) ? given that the notch is > 10s microns across and deep ? What are the tolerances here, and how uniform does it need to be ? How can ‘overspray’ on the outside of the fibre of the graphene and silver be addressed ? If it could be fabricated, what are the consequences if estimated fabrication and manufacturing tolerances are included in this model ?

In terms of using this sensor in an analyte, has the effect of the analytes viscosity, surface tension been considered as I expect that the analyte would need to be able to penetrate these small notches ? If the analyte did penetrate how would it be removed/ flushed to allow new samples to be tested ?

How would the end of the fiber be sealed so that liquid does not penetrate inside the fiber ?

Author Response

(The authors gave the same response as above.)

Round 2

Reviewer 3 Report

I have re-reviewed the manuscript and appreciate the thoroughness that the authors have demonstrated in addressing and answering my questions. 

Grammar and english have improved, however it still requires professional editing.